# Rice PIN Auxin Efflux Carriers Modulate the Nitrogen Response in a Changing Nitrogen Growth Environment

**DOI:** 10.3390/ijms22063243

**Published:** 2021-03-23

**Authors:** Yun-Shil Gho, Min-Yeong Song, Do-Young Bae, Heebak Choi, Ki-Hong Jung

**Affiliations:** Graduate School of Biotechnology & Crop Biotech Institute, Kyung Hee University, Yongin 17104, Korea; koyoong@khu.ac.kr (Y.-S.G.); min2187@khu.ac.kr (M.-Y.S.); dybae323@khu.ac.kr (D.-Y.B.); backy28@khu.ac.kr (H.C.)

**Keywords:** rice, auxin, auxin efflux carrier, ammonium-dependent response, ammonium assimilation, *ospin1b* mutant

## Abstract

Auxins play an essential role in regulating plant growth and adaptation to abiotic stresses, such as nutrient stress. Our current understanding of auxins is based almost entirely on the results of research on the eudicot *Arabidopsis thaliana*, however, the role of the rice PIN-FORMED (PIN) auxin efflux carriers in the regulation of the ammonium-dependent response remains elusive. Here, we analyzed the expression patterns in various organs/tissues and the ammonium-dependent response of rice *PIN*-family genes (*OsPIN* genes) via qRT–PCR, and attempted to elucidate the relationship between nitrogen (N) utilization and auxin transporters. To investigate auxin distribution under ammonium-dependent response after N deficiency in rice roots, we used DR5::VENUS reporter lines that retained a highly active synthetic auxin response. Subsequently, we confirmed that ammonium supplementation reduced the DR5::VENUS signal compared with that observed in the N-deficient condition. These results are consistent with the decreased expression patterns of almost all *OsPIN* genes in the presence of the ammonium-dependent response to N deficiency. Furthermore, the *ospin1b* mutant showed an insensitive phenotype in the ammonium-dependent response to N deficiency and disturbances in the regulation of several N-assimilation genes. These molecular and physiological findings suggest that auxin is involved in the ammonium assimilation process of rice, which is a model crop plant.

## 1. Introduction

Auxins play an essential role in regulating plant growth and adaptation to abiotic stresses, and their spatial distribution depends largely on the polar localization of the PIN-FORMED (PIN) auxin efflux carrier family. The PIN family plays a role in directing intercellular auxin flow and mediating the auxin-regulated developmental processes [1]. *Arabidopsis* carries eight members of this protein family, and the AtPIN proteins are divided into two types: one has a long hydrophilic loop (“long PIN proteins”) and the other contains a short hydrophilic loop (“short PIN proteins”). Both types have been reported to participate in various developmental processes [2,3,4]. AtPIN1–4 and AtPIN7, with the long hydrophilic loop, are located in the plasma membrane and are involved in root development and apical shoot establishment [5,6]. AtPIN5 and AtPIN8 are localized in the endoplasmic reticulum and play roles in the homeostasis of intracellular auxins [7,8]. Interestingly, AtPIN6 is localized in both plasma membranes and the endoplasmic reticulum [9]. Homologs are also found in rice, which has 12 potential PIN family members, including: four *PIN1*s and one *PIN2*, which belong to the long PINs; three *PIN5*s and one *PIN8*, which belong to the short PINs; and three monocot-specific PINs (*PIN9*, *PIN10a*, and *PIN10b*) [10]. The rice *PIN* genes are involved in hormone signaling and abiotic stress responses [11,12,13]. However, our current knowledge of the auxin signaling pathway associated with the *PIN* family is almost entirely based on the results of research on the *PIN*-family genes of the eudicot *Arabidopsis thaliana*. Research on *PINs* as auxin transporters related to nutrients, especially N, is scarce. 

N is one of the major nutrients in plants and is essential for normal growth. The N supply for plant growth is mainly obtained from fertilizer supplementation; however, nitrogen use efficiency is quite low, and excess nitrogen causes environmental pollution, including water and soil contamination and an increase in production costs. Plants absorb N from the soil in the form of nitrate (NO_3_^−^) and ammonium (NH_4_^+^) via the roots using ammonium transporters (AMTs). There are at least 12 putative *OsAMT*-like genes in rice further divided to five classes (*OsAMT1-5*) [14]. OsAMT1 class members are characterized as high affinity transport systems (HAT), while the other four classes are characterized as low affinity transport systems (LAT) [15,16]. *OsAMT1;1* is constitutively expressed in the leaves but is stimulated in the roots under ammonium supply. The expression of *OsAMT1;2* is both root-specific and ammonium-inducible, whereas the expression of *OsAMT1;3* is root-specific but is suppressed under ammonium supply [17,18]. The NH4^+^ is toxic and so it needs to be rapidly assimilated by the Gln synthetase (GS)/Glu synthase (GOGAT) pathway; this process requires energy, a reductant, and C skeletons in the form of 2-oxoglutarate (2-OG) [19,20]. Rice has two types of GS: three homologous genes for cytosolic GS1 (*OsGS1;1*, *OsGS1;2* and *OsGS1;3*) and one chloroplastic GS2 gene (*OsGS2*), which carries out the first step in inorganic N incorporation into amino acids by transferring ammonium to Glu to form Gln [21]. *OsGS1;1* was found to be highly expressed in all tested organs in leaf blades, while *OsGS1;2* and *OsGS1;3* were expressed mainly in the roots and spikelet, respectively [22]. *OsGS2* is the major GS isoform and is located in mesophyll cell chloroplasts. The GOGAT performs the second step in the GS/GOGAT cycle and catalyzes the reductant-dependent conversion of Gln and 2-OG to two molecules of Glu. Rice has two NADH-types (two GOGATs and one Fd-GOGAT) [23]. *OsNADH-GOGAT1* is expressed in the epidermis of the root after the supply of NH4^+^, and is primarily involved in ammonium assimilation in the root [24]. Recently, it has also been reported that dual activation of *OsAMT1;2* and *OsGOGAT1* improves crop agriculture traits, providing better grain filling without yield penalties in paddy conditions and better grain yield and N content when growing plants under N restriction [25]. Several studies aimed at improving the N-use efficiency have been reported, but the practical application of this knowledge remains fairly limited.

Recently, auxins have been shown to play important roles in plant adaptation in response to the fluctuating availability of usable N [26]. Moreover, *OsPIN1b* has been shown to be involved in seminal root elongation by regulating root apical meristem activity under low-nutrient conditions [27]. However, little is known about the role of *OsPIN*s at the transcriptional level under N-supplementation conditions. Therefore, the functional involvement of the *PIN*-family genes in N utilization is a particularly interesting topic for further study.

In the present study, we examined the functional roles of rice PIN family members through a diverse spatial expression analysis using qRT–PCR and a meta-expression analysis of data from 983 Affymetrix arrays [28]. We performed a time-course experiment to examine the expression patterns of the *OsPIN*s gene in response to N availability. After supplementation with N, following long-term N deprivation, almost all *OsPIN-*family genes were substantially downregulated. These results are consistent with the decreased expression pattern of the DR5::VENUS reporter lines. The DR5::VENUS reporter lines retain a highly active synthetic auxin response element and can reflect in vivo auxin levels [29]. Furthermore, we identified the insensitive phenotypes in response to N supplementation in the *ospin1b* mutant and discussed the relationship between the N stress response and the *OsPIN1b*-mediated ammonium assimilation signaling pathway.

## 2. Results

### 2.1. Identification and Functional Assignment of Rice PIN Family Members in Various Tissues/Organs

Recently, Balzan (2014) first isolated 12 *PIN*-family genes in rice. However, biological information on the family members has not been particularly well described in rice. In the present study, we attempted to assign biological functions to the *PIN* genes in rice. To do this, we first carried out a comparative phylogenic analysis of 12 *PIN* genes from rice, 12 *PIN* genes from maize, and eight *PIN* genes from *Arabidopsis*. We identified eight subgroups (Appendix A); among them, four were shared with *Arabidopsis*, a model eudicot plant, two were monocot-specific subgroups and two were eudicot-specific subgroups. Therefore, we identified six subgroups in the rice *PIN* (*OsPIN*) family. Subgroups 1, 2, 3, and 4 were conserved in both dicots and monocots, whereas Subgroups 5 and 6 existed only in monocots. Subgroup 1 comprised four *OsPIN* genes (*LOC_Os02g50960/OsPIN1b*, *LOC_Os06g12610/OsPIN1a*, *LOC_Os11g04190/OsPIN1c*, and *LOC_Os12g04000/OsPIN1d*), whereas Subgroup 2 contained one *OsPIN* gene (*LOC_Os06g44970/OsPIN2*), Subgroup 3 contained three genes (*LOC_Os01g69070/OsPIN5a*, *LOC_Os08g41720/OsPIN5b*, and *LOC_Os09g32770/OsPIN5c*), Subgroup 4 encompassed one gene (*LOC_Os01g51780/OsPIN8*), subgroup 5 comprised one gene (*LOC_Os01g58860**/OsPIN9*), and Subgroup 6 had two genes (*LOC_Os01g45550**/OsPIN10a* and *LOC_Os05g50140**/OsPIN10b*). The detailed information of these findings is summarized in Appendix A. Although the phylogenic analysis suggests the presence of functional similarity among the rice, maize, and *Arabidopsis* PIN gene products clustered in the same subgroup of the phylogenetic tree, it is not clear whether their biological functions are conserved between monocot and dicot plants. To identify the functional roles of *OsPIN* genes, we carried out a qRT–PCR analysis using leaf blade, leaf sheath, roots, mature flowers, mature seeds (seed), and young panicle tissues (Figure 1). *OsPIN1a*, *OsPIN1b*, *OsPIN2*, *OsPIN9*, and *OsPIN10b* exhibited the highest expression level in roots. In Subgroup 4, *OsPIN5a* was significantly expressed in leaf blades, and *OsPIN5a* and *OsPIN5c* showed a reproductive-tissue-preferred expression pattern. The expression patterns of *OsPIN1c* and *OsPIN1d* from *PIN* Subgroup 1 and *OsPIN10a* from Subgroup 6 suggested their roles in young panicle development; however, the overall expression of *OsPIN8* was very low. The expression patterns of *OsPIN* genes determined by qRT–PCR suggest that they are useful indicators with respect to the functions of individual genes.

### 2.2. Morphological Appearance of Rice Seedlings under Conditions of Nitrogen Depletion and Ammonium Supplementation

To explore the responses of rice seedlings under conditions of N depletion and ammonium supplementation, pre-germinated seedlings were grown hydroponically for 14 days in distilled water, and further grown in N-deficient Yoshida solution (0 mM (NH_4_)_2_SO_4_) for 3 days. Half of these plants were transferred to an N-deficient solution, and the other half remained in the N-sufficient (0.5 mM (NH_4_)_2_SO_4_) solution for up to 48 h. After completion of these treatments, we examined the changes in the morphological appearance of the seedlings and measured quantitative changes in all seedlings. For quantitative comparisons, the lengths of the roots and shoots were measured in all individual seedlings, and the mean value of each variable from 10 seedlings per treatment was determined from seedlings grown under both the N-deficient and N-supplemented conditions. Although there was no statistically significant difference, the root and shoot lengths under the N-supplemented condition was slightly greater than those analyzed under the N-deficient condition (Figure 2a,b). We examined changes in ammonium content, and it was confirmed that the ammonium content was significantly enhanced in both shoots and roots treated with ammonium for 48 h compared to those under N-deficient conditions (Figure 2c). Our observations revealed that the N source was particularly important with respect to the stem and root growth of rice seedlings (Figure 2).

### 2.3. OsPIN-Family Genes Are Closely Associated with Ammonium Treatment in Rice

It has been shown that auxins play important roles in plant adaptation to N availability under climate change conditions [26]. To further test the relationship between the N response and *OsPIN* genes encoding auxin transporter in rice roots, we performed a time-course analysis of the gene expression patterns in response to N availability. The plants were grown in either water or N-depleted hydroponic culture medium for 17 days. Half of the seedlings continued to be cultured under the same conditions, whereas the remaining seedlings were transferred to N-sufficient medium and incubated for up to 7 h. In total, three-time points were selected to test for short- and long-term responses to N deprivation and the effects of N supplementation on N-starved plants. To evaluate the N response of the tested samples, we first checked the expression patterns of two marker genes for the nitrogen response, i.e., *OsGS1;2* and *OsNADH-GOGAT1* [23] (Figure 3) and found that peaks of expression of the two marker genes were observed at 3 h following N supplementation, after which their expression levels gradually decreased. The time course of the expression patterns of the *OsPIN* genes was then monitored under the same conditions. Our data showed that the expression of five *OsPIN* genes (*OsPIN1a*, *OsPIN1b*, *OsPIN2*, *OsPIN9*, and *OsPIN10a*) with root-preferred expression patterns was lower under the N-sufficient condition than it was under the N-deficient condition. *OsPIN1a* and *OsPIN9* were responsive to the N supplementation at all time points tested. However, *OsPIN1b* and *OsPIN2* seemed to be involved in the early N response, whereas *OsPIN10a* appeared to function a little later in the roots under N supplementation. Interestingly, the expression of *OsPIN5c*, which exhibits a root-preferred expression pattern, was only higher under the N-sufficient condition compared with the N-deficient condition. Although *OsPIN5a*, which exhibits a leaf-sheath-preferred expression pattern, was only weakly expressed in the roots, its expression was regulated by N supplementation at all time points tested. Therefore, *OsPIN5a* might play an important role in the transfer of auxins from the roots to the shoots under N supplementation. Taken together, these results indicate that the function of the *PIN* family genes is closely associated with the N response in rice.

### 2.4. Ammonium Resupply Decreases Auxin Response in Root Tips and Lateral Roots via the DR5::VENUS System

Auxins might be involved in the regulation of root development in response to N supply. To investigate the interaction between auxin accumulation and root architecture changes for the duration of the N supply, we used the DR5::VENUS reporter line, which shows a highly active synthetic auxin response and, thus, can reflect in vivo auxin levels [29]. The DR5::VENUS reporter line was transferred to an N-deficient solution (0 mM N) for 3 days after incubation of pre-germinated seedlings for 14 days in a hydroponic culture medium with distilled water. These conditions represented 17 days of N-deprivation treatment. Half of these plants were incubated in an ammonium-supplemented solution (0.5 mM (NH_4_)_2_SO_4_) for up to 48 h. Compared with the N-deficient condition (0 mM (NH_4_)_2_SO_4_), we confirmed that ammonium supplementation for 48 h decreased the DR5-VENUS signal in the root tips and in the lateral root initiation zone (Figure 4). These results were consistent with the decreased expression pattern of the *OsPIN*-family genes in the presence of an ammonium-dependent response following N deficiency (Figure 3). 

### 2.5. The Ospin1b Mutant (3A-04335) Displayed No Response to Ammonium Supplementation after Long-Term N Deficiency

To analyze the functional significance of *OsPIN*-family genes in response to ammonium supply after long-term N deficiency, we used an *ospin1b* mutant that has been reported previously [27]. We grew *ospin1b* and WT plants for 10 days after germination on Murashige and Skoog (MS) media, and the seedlings were incubated hydroponically for 14 days in distilled water and further grown in N-deficient Yoshida solution (0 mM (NH_4_)_2_SO_4_) for 3 days. We confirmed that the *ospin1b* mutant has a shorter root phenotype compared with the WT plants, as reported previously (Figure 5a). Half of these plants were transferred to an N-deficient solution, and the remaining plants were transferred to an N-sufficient solution (0.5 mM (NH_4_)_2_SO_4_). After exposure to the N-sufficient hydroponic medium for 48 h, the root and shoot lengths and weight of the WT under N-supplemented conditions were increased significantly more than those analyzed under N-deficient conditions, whereas the *ospin1b* mutant exhibited an insensitive phenotype in response to N-supplementation (Figure 5a–c). We also measured the leaf color and chlorophyll content between the WT and *ospin1b* mutants under N-supplementation and N-deficiency conditions. We then identified that the chlorophyll content of the WT was increased when nitrogen was resupplied after long-term N-starvation, but the N-supplementation effect did not occur significantly in the *ospin1b* mutant (Figure 5d,e). In addition, we found that there was no significant difference in the ammonium content of the *ospin1b* mutants between N-supplementation and N-deficiency conditions (Figure 6). These results indicate that the *ospin1b* mutant is insensitive to N-supplementation after long-term N-starvation. 

### 2.6. Mutation of OsPIN1b Alters the Expression Pattern of Ammonium-Response-Related Genes

To investigate the effect of a mutation in the *OsPIN1b* gene on the expression of genes involved in the N-signaling pathway, qRT–PCR was carried out using root samples of WT and *ospin1b* plants grown under N-deprivation conditions (0 mM (NH_4_)_2_SO_4_) for 17 days and 3 h, or in N-supplemented medium (0.5 mM (NH_4_)_2_SO_4_) for 3 h, after N-deprivation conditions for 17 days. To further determine how the *ospin1b* mutation altered the root development mediated by ammonium-dependent response genes and root-growth-related genes, we analyzed the expression pattern of *OsAMT1;1–3*, *OsGS1;2*, *OsNADH-GOGAT1*, *OsGDH2*, *OsAS2*, and *OsGLU3* as the ammonium-dependent response genes [23,30,31] (Figure 7). As expected, five genes (*OsAMT1;1*, *OsAMT1;2*, *OsGS1;2*, *OsNADH-GOGAT1*, and *OsGDH2*) were upregulated and two genes (*OsAMT1;3* and *OsAS2*) were downregulated when placed under the N-supplementation condition for 3 h after long-term N-deprivation treatment in WT plants. Interestingly, although the expression of *OsAMT1;2* and *OsGDH2* was increased by ammonium supplementation following N deficiency, their expression change was significantly decreased in the *ospin1b* mutant compared with the WT plants. The expression of *OsNADH-GOGAT1* under ammonium supply following N deficiency was increased in the *ospin1b* mutant compared with the WT plants. However, the expression of *OsGS1;2*, *OsAMT1;1*, *OsAMT1;3*, and *OsAS2* was not significantly different between the *ospin1b* mutant and WT plants. Taken together, our results suggest that *OsAMT1;2* and *OsGDH2* are responsible for the ammonium-dependent response in the *ospin1b* mutant.

In the case of *OsGLU3*, which is related to root development and functions in the synthesis of cellulose when nitrogen is deficient, its expression was decreased when ammonium was administered under nitrogen-deficient conditions for a long period of time in WT plants. In the *ospin1b* mutant, the expression of *OsGLU3* not only dramatically decreased in the N-deprivation condition, but also significantly reduced when ammonium was supplied (Figure 7). This result indicates that *OsGLU3* is regulated by *OsPIN1b*, regardless of N depletion or N supplementation. 

## 3. Discussion

### 3.1. Roles of PIN-Family Genes in Rice

Although auxin plays an essential role in regulating both plant growth and tolerance to abiotic stresses, research on the relationship between nitrogen utilization efficiency and auxins is very limited. In this study, we first tried to identify the functional roles of the *PIN* (auxin efflux carrier) family of genes in rice, through a qRT–PCR analysis in various organs/tissues and in the presence of the ammonium-dependent response. Rice carries 12 PIN family members, and a phylogenetic analysis classified these members into six subgroups [32]. Our results indicated that Subgroups 1, 2, 3, and 4 were auxin efflux carriers that existed in both dicots and monocots, whereas Subgroups 5 and 6 were auxin efflux carriers that existed only in monocots. As indicated, *OsPIN10a* and *OsPIN10b* of Subgroup 6 and *OsPIN9* of Subgroup 5 might be monocot specific, as they are also conserved in maize and rice (Appendix A) [10,33]. It has been shown that *AtPIN1* is essential for the initiation and development of lateral root primordia in *Arabidopsis* [34]. Unlike *AtPIN1*, which functions as the sole *PIN*-family member in the *Arabidopsis* genome, the rice *PIN1* family contains four homologs, namely, *OsPIN1a*, *1b*, *1c*, and *1d*. Wang et al., (2004) tried to identify the functional roles of *OsPIN* genes through a spatial/temporal expression analysis and organization of the protein structures, and confirmed that expression of all *OsPIN* genes, except *OsPIN2* and *OsPIN9*, was enhanced following treatment with 10 μM indole-3-acetic acid (IAA) for 7 days. Among the nine *OsPIN* genes, *OsPIN1a* and *OsPIN1b* are only induced by IAA, and *OsPIN1b* of them is expressed in the root cap where the distal auxin maximum is created [35]. Recently, it has been shown that *OsPIN1b* is involved in seminal root elongation by regulating root apical meristem activity under low-nutrient conditions. It was also confirmed that the [^3^H]IAA transport activity and auxin concentration decreased significantly at the root tip of the *ospin1b* knockout mutant [27]. Moreover, we confirmed that the *ospin1b* mutant displayed markedly shorter roots and a lower number of crown roots (Figure 5). These results indicate that *OsPIN1**b* is an auxin transporter that regulates the development of the roots in both normal and nutrient-limited conditions [35,36] (Figure 1 and Figure 5).

### 3.2. Auxin Is Involved in the Ammonium-Dependent Response Mediated by Rice PIN-Family Genes Encoding Auxin Efflux Carriers

Ammonium is the major source of inorganic nitrogen for plants. Ammonium promotes plant growth at low external supplies, while high ammonium supplies cause toxicity in plants. Recently, auxins have been shown to play important roles in plant adaptation to the fluctuating availability of usable nitrogen [26]. In *Arabidopsis*, the auxin-resistant mutants *aux1*, *axr1*, and *axr2* were shown to be insensitive to NH_4_^+^-mediated root growth inhibition [37], and shoot-supplied NH_4_^+^ inhibited lateral root emergence by interfering with AUX1, an auxin influx transporter, from shoots to roots [38]. In addition, NH_4_^+^ treatment altered the expression levels of several auxin signaling genes (*OsIAA24* (LOC_Os07g08460), *OsIAA19* (LOC_Os05g48590), *OsGH3-13* (LOC_Os11g32510), and *OsIAA10* (LOC_Os02g57250)) and auxin transporters (*OsPIN5a*, *OsPIN5b*, and *OsPIN5C*) and delayed the rice root tip response to gravity changes [39]. These findings suggest that auxin is tightly associated with NH_4_^+^-mediated root growth. Conversely, the molecular mechanism of auxin and NH_4_^+^ interaction remains elusive in rice. Our results also indicate that PIN expression is repressed by NH_4_^+^ supply in rice roots, and that the auxin response is decreased in the rice root tips and in the lateral root initiation zone under the ammonium-dependent response following N deficiency, as assessed using DR5-VENUS lines (Figure 4). The five *OsPIN* genes that are localized to the plasma membrane and exhibit root-preferred expression patterns [40,41], i.e., *OsPIN1a*, *OsPIN1b*, *OsPIN2*, *OsPIN9*, and *OsPIN10a*, were negatively regulated by the NH_4_^+^-dependent response (Figure 3). Conversely, *OsPIN5s*, which has been reported to localize to the endoplasmic reticulum, showed a different pattern among the three members [10] (Figure 1). *OsPIN5c*, which is highly expressed in leaf blades, responded negatively to the NH_4_^+^-dependent response, whereas *OsPIN5a* and *OsPIN5b*, which are highly expressed in seeds and flowers, responded positively to the NH_4_^+^-dependent response (Figure 3). 

Root gravitropism is affected by many environmental stresses including salinity, drought, and nutritional deficiencies [42,43,44]. Recently, it was reported that excessive ammonium (NH_4_^+^) not only inhibits root elongation and lateral root formation, but also accompanies the loss of root gravitropism in *Arabidopsis*. Zou et al. (2012) showed that excessive NH_4_^+^ treatment significantly decreased the expression of the β-glucuronidase (GUS) signal of DR5::GUS as the auxin-responsible reporter in root tip cells [45]. Our results also show that the DR5::VENUS signal and the expression levels of most of the *PIN* genes were reduced when ammonium was given after a prolonged period of nitrogen deficiency (Figure 3 and Figure 4). *AtPIN2*, that is predominantly found with apical polarity in the root epidermis and lateral root cap and basal polarity in the cortex, is a major component for basipetal auxin transport, with an involvement in root gravitropism [46,47,48]. Although we did not observe a change in root gravity during NH_4_^+^ treatment, it was confirmed that the expression of the *OsPIN2* gene, in addition to the homology of *AtPIN2*—which was previously known to be involved in root gravitropism—decreased during NH_4_^+^ treatment after long-term N deficiency (Figure 3). This suggests that *OsPIN2* might be involved in root gravitropism under the NH_4_^+^ response.

In addition, we identified the phenotype of the *ospin1b* mutant after treatment with ammonium following long-term N deprivation. The biomass and ammonium contents of the roots and shoots were increased in WT plants in response to ammonium supplementation after long-term N deprivation, whereas no response was observed in the *ospin1b* mutant under the same conditions (Figure 5 and Figure 6). Ammonium nutrition is known to affect chlorophyll content in rice [49]. Our results were consistent with the fact that the WT increased the total chlorophyll content when ammonium was supplied after prolonged depletion of all nitrogen sources, whereas the mutants did not differ in chlorophyll content (Figure 5d,e). Low auxin levels are known to increase sugar levels, thereby inhibiting the expression of photosynthetic genes and the production of chlorophyll [50]. Moreover, it has been reported that the leaves of the *OsPIN5*-overexpressing transgenic plant have increase chlorophyll levels [10]. Our results also suggest that nitrogen supply increases chlorophyll production, while low auxin levels inhibit chlorophyll production. Therefore, these results reveal that *OsPIN1b* might be involved in the growth of rice plants due to its association with the ammonium response after long-term N deprivation.

### 3.3. OsPIN1b Is Involved in Ammonium Assimilation

The *ospin1b* mutant displayed suppressed root development compared with the WT plants as a consequence of the ammonium response after long-term N deprivation. To examine whether the mutation of the *OsPIN1b* gene is involved in the N-signaling pathway, the expression of the N-signaling pathway-related genes was analyzed in the *ospin1b* mutant. Two ammonium-assimilation-related genes (*OsAMT1;2* and *OsGDH2*) were significantly altered in *ospin1b*. Ammonium assimilation is an important process of the ammonium-induced physiological response in plants. It has been suggested that the ammonium-use efficiency of plants is related to their ability to assimilate ammonium. Once ammonium has entered plant cells via the ammonium transporter 1 (AMT1) [15,17], the ammonium is rapidly assimilated into glutamine and glutamate through the glutamine synthase/glutamate synthase (GS-GOGAT) cycle [51]. In addition to the GS-GOGAT cycle, an alternative pathway of ammonium assimilation is driven by the NADH-dependent glutamate dehydrogenase (GDH), which synthesizes glutamate in the cytoplasm using ammonium and 2-oxoglutarate and is upregulated by the supply of ammonium [52]. Rice *INDETERMINATE DOMAIN 10* (*OsIDD10*), a NH_4_^+^ signaling transcription factor, was reported to regulate the expression of NH_4_^+^ uptake and N-assimilation genes [39]. *Osidd10* mutants exhibited an ammonium-hypersensitive root growth defect [53]. Furthermore, among the ammonium uptake and assimilation genes, *OsAMT1;2* and *OsGDH2* were shown to be significantly dependent on the regulation of *OsIDD10* expression for ammonium-mediated induction processes [53]. The expression levels of both *OsAMT1;2* and *OsGDH2* were significantly lower in the *ospin1b* mutant compared with WT plants for the ammonium response after long-term N deprivation (Figure 7). This suggests that *OsAMT1;2* and *OsGDH2* in the ammonium assimilation pathway can be regulated by *OsPIN1b*. However, the clear result of mutual regulation needs to be confirmed in future studies. *OsGLU3*, which encodes a beta-1,4-endoglucanase, affects cellulose synthesis for root elongation in rice [31]. *Osglu3-1* has been reported to have a reduced content of crystalline cellulose in its root cell walls, a shorter root cell length, and a slightly smaller root meristem [54]. Interestingly, *OsGLU3* expression was dramatically reduced in the *ospin1b* mutant (Figure 7), and a shorter root length was observed in the *ospin1b* mutant compared with WT plants, implying that *OsPIN1b* might be involved in the root cell elongation by positively regulating the expression of *OsGLU3*. Based on the current study and previous studies, we propose a simple model to explain the relationship between *OsPIN1b* and N-assimilation in rice under the NH_4_^+^-dependent response (Figure 8). Taken together, our findings provide molecular and physiological evidence that the *PIN* family is involved in the ammonium assimilation process of rice, which is a model crop. This study is intended to be helpful for future research pertaining to the improvement of the nitrogen use efficiency associated with *OsPIN1b*.

## 4. Conclusions

In summary, by qRT-PCR analysis of *OsPINs* and the expression analysis of the DR5::VENUS reporter line, we have shown that ammonium supply after long-term N starvation could lead to reduced auxin distribution. Furthermore, the *ospin1b* mutant showed an insensitive phenotype under the ammonium-dependent response after long-term N deficiency and regulated several genes in the N-assimilation pathway. These molecular and physiological findings suggest that auxin is involved in the ammonium assimilation process of rice. We expect that this study will be helpful in improving nitrogen use efficiency in crop plants, acting as a basis for future studies.

## 5. Materials and Methods

### 5.1. Multiple Sequence Alignment and Phylogenetic Analysis

To perform a phylogenomic analysis of PIN-family proteins in rice, *Arabidopsis*, and maize, we collected 12 family members from the Rice Genome Annotation Project (http://rice.plantbiology.msu.edu/accessed on 15 February 2021), as well as 8 *Arabidopsis* and 11 maize family members from GreenPhyl v5 (http://www.greenphyl.org/cgi-bin/index.cgi accessed on 15 February 2021) (Appendix A). The phylogenetic analysis used MEGA7 under the following parameters: neighbor joining, complete deletion, and bootstrap with 500 replicates (Appendix A), as described previously [55,56]. We also generated a phylogenic tree for rice PINs, to integrate meta-expression data from data from 983 Affymetrix arrays (Figure 1). The multiple sequence alignment of these proteins was conducted using the ClustalX program [57].

### 5.2. Meta-Analysis of Organ-Specific Expression Profiles

The integration of transcriptomes into a phylogenetic context can drive more effective experimental strategies for further functional analysis [58]. Therefore, we used a meta-analysis of the expression profiles of *OsPIN*-family genes in six tissues/organs based on the data from 983 Affymetrix arrays, downloaded from the NCBI gene expression omnibus (GEO, http://www.ncbi.nlm.nih.gov/geo/ accessed on 15 February 2021) [28]. We then uploaded the log_2_ normalized intensity data in tab-delimited text format into the Multi Experiment Viewer (MEV, http://mev.tm4.org/#/welcome accessed on 15 February 2021) and illustrated these data using heat maps (Figure 1).

### 5.3. Plant Material and Stress Treatment

Rice (*Oryza sativa L. cv. Dongjin*) seeds were germinated on MS medium under controlled conditions of 28 °C day/25 °C night temperatures, 8 h light/16 h dark cycle, and 78% relative humidity, after sterilization with 50% (*w/v*) commercial bleach for 30 min with gentle shaking, as described previously [59,60]. For the anatomical expression analysis, roots, leaf sheaths, leaf blades, panicles before heading, flowers at the heading stage, and seeds at 10 and 15 days after pollination were harvested to extract total RNA. For the ammonium response analysis in rice after long-term N starvation, we used rice *cv. Dongjin*, and the DR5-GFP-expressing line and the T-DNA insertional mutant of the *OsPIN1b* gene (3A-04335) were germinated on MS medium at 28 °C for 10 days. The seedlings were incubated hydroponically for 14 days in distilled water and further grown in N-deficient Yoshida solution (0 mM (NH_4_)_2_SO_4_) for 3 days. Half of these plants were transferred to an N-deficient solution, and the remaining plants were transferred to an N-sufficient solution (0.5 mM (NH_4_)_2_SO_4_). The Yoshida medium consisted of 0.3 mM NaH_2_PO_4_, 0.5 mM K_2_SO_4_, 1.0 mM CaCl_2_, 1.6 mM MgSO_4_, 0.17 mM NaSiO_3_, 50 µm Fe-EDTA, 0.06 µM (NH_4_)_6_Mo_7_O_24_, 15 µMH_3_BO_3_, 8 µM MnCl_2_, 0.12 µM CuSO_4_, 0.12 µM ZnSO_4_, 29µ MFeCl_3_, and 40.5µM citric acid, without (NH_4_)_2_SO_4_ [61]. The pH of the culture solution was adjusted to 5.5 using 1 M NaOH.

### 5.4. RNA extracTion and Real-Time PCR

The anatomical sample and ammonium response sample in rice were frozen in liquid nitrogen and ground with a Tissue Lyser II (Qiagen, Hilden, Germany). The RNA from both was extracted using the RNAiso Plus Kit, according to the manufacturer’s protocol (Takara Bio, Kyoto, Japan). To determine the relative gene expression patterns by qRT–PCR, we used a primer set for rice ubiquitin 5 (*OsUbi5*, *LOC_Os01g22490*) [62]. For real-time PCR, the cycling conditions were 95 °C for 9 s, 57 °C for 10 s, and 72 °C for 20 s. All experiments were repeated three times using the same control primer sets and different biological replicates. All primers used in the qPCR analysis are described in Appendix A.

### 5.5. Genotyping of T-DNA Insertional Lines for the OsPIN1b Gene

The *ospin1b* mutant, PFG_3A-04335, was obtained from RiceGE (http://signal.salk.edu/cgi-bin/RiceGE accessed on 15 February 2021). The functional role of this gene has been reported previously [15]. To confirm the homozygous mutants by T-DNA insertion in the *OsPIN1b* gene, we genotyped the *OsPIN1b*-segregating population using a gene-specific primer set in a PCR assay. We also conducted an expression analysis of *OsPIN1b* in *ospin1b* mutants by qRT–PCR. All experiments were repeated three times using the same control primer sets and different biological replicates. All primer sequences used in the PCR and qPCR experiments are listed in Appendix A.

### 5.6. IAA Distribution Assay Using the DR5:GFP System (DR5::VENUS line) in Response to Ammonium

To analyze the IAA distribution in rice under the ammonium response, we used DR5::VENUS reporter lines. First, the seeds of DR5::VENUS reporter lines were sterilized using 50% bleach for 25 min with gentle shaking and then washed 4–5 times with sterile triple-distilled water. Subsequently, the reporter lines were germinated on MS solid medium and the seedlings were grown vertically for 10 days after germination. The seedlings were transferred to an N-deficient solution (0 mM (NH_4_)_2_SO_4_) for 3 days after 14 days in a hydroponic culture medium in distilled water. Half of the seedlings were grown under the same conditions with an N-deficient solution (0 mM (NH_4_)_2_SO_4_) for 48 h, whereas the other half were transferred to an ammonium-supplemented solution (0.5 mM (NH_4_)_2_SO_4_)) and incubated for up to 48 h. The tissues were photographed using a laser-scanning confocal microscope (Carl Zeiss, Jena, Germany).

### 5.7. Measurement of Chlorophyll Content

For chlorophyll measurements, approximately 50 mg samples of fresh leaves were gathered from different N solutions (N-deficient (0 mM (NH_4_)_2_SO_4_) and N-sufficient (0.5 mM (NH_4_)_2_SO_4_) solution) for 48 h after 17 days of N-deprivation treatment. The anatomical sample and ammonium response sample in rice were frozen in liquid nitrogen, weighed, and ground with a Tissue Lyser II (Qiagen, Hilden, Germany). Chlorophyll was extracted with 80% aqueous acetone at 4 °C under darkness. The extract was measured spectrophotometrically at 645 and 663 nm. A total amount of 80% aqueous acetone was used as a blank control. The chlorophyll content was calculated as described by [63].

### 5.8. Measurement of Ammonium

Plants were treated with ammonium responses from different N solutions (–N, N-deficient solution; +N, N-sufficient solution) for 3 h after 17 days of N-deprivation treatment and soaked for 60 min in freshwater to remove extracellular NH4+. Samples were ground in liquid nitrogen after fresh-weight measurement (approximately 100 mg), extracted in 1 mL of 100 mM HCl, and subsequently, 0.5 mL of chloroform was added. After centrifugation at 12,000× *g* for 10 min at 4 °C, the supernatant was used for NH4 + concentration determination by phenol hypochlorite assay (Berthelot reaction). The following three solutions were added in the 200 μL of tissue extract; 200 μL of 11 mM phenol in 95% (*v*/*v*) ethanol;) 200 μL of 1.7 mM sodium nitroprusside; 500 μL of the oxidizing solution containing 100 mL of 0.68 M trisodium citrate in 0.25 M NaOH with 25 mL of sodium hypochlorite. The sample was incubated at 25 °C for 90 min and absorbance was measured at 630 nm. Three biological replicates of each of the three samples were analyzed. For each sample, two technical measures were recorded to check the accuracy, and the average of both measures was used. We conducted ammonium measurements according to [64] and [65].

## Figures and Tables

**Figure 1 ijms-22-03243-f001:**
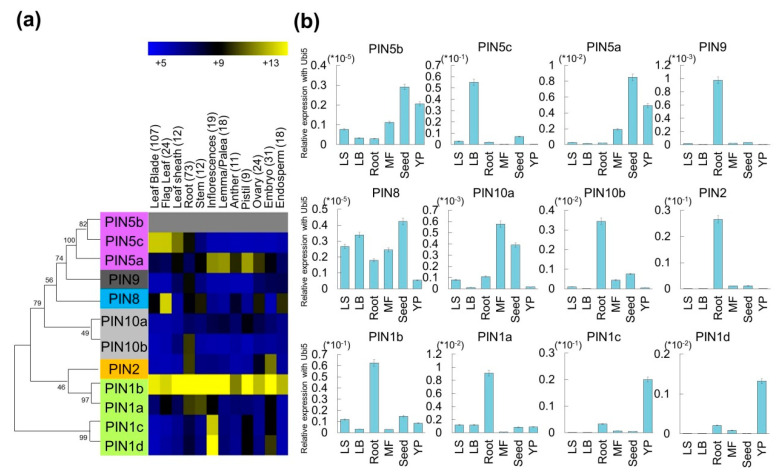
Heatmap expression analysis of *OsPIN*-family genes using meta-anatomical expression data, and validation of the expression patterns using qRT–PCR. (**a**) Affymetrix expression data consisting of eight tissues/organs were used to create the heatmap via the MeV software. Blue, low levels of expression; yellow, high levels of expression. (**b**) qRT-PCR was performed in six tissues/organs (leaf sheaths, leaf blades, roots, mature flowers, seeds, and young panicles). X-axis, tissues/organs used for qRT–PCR analysis; Y-axis, relative expression level to that of the rice ubiquitin 5 gene (*OsUbi5*, *LOC_Os01g22490*). All experiments were repeated three times using four biological replicates.

**Figure 2 ijms-22-03243-f002:**
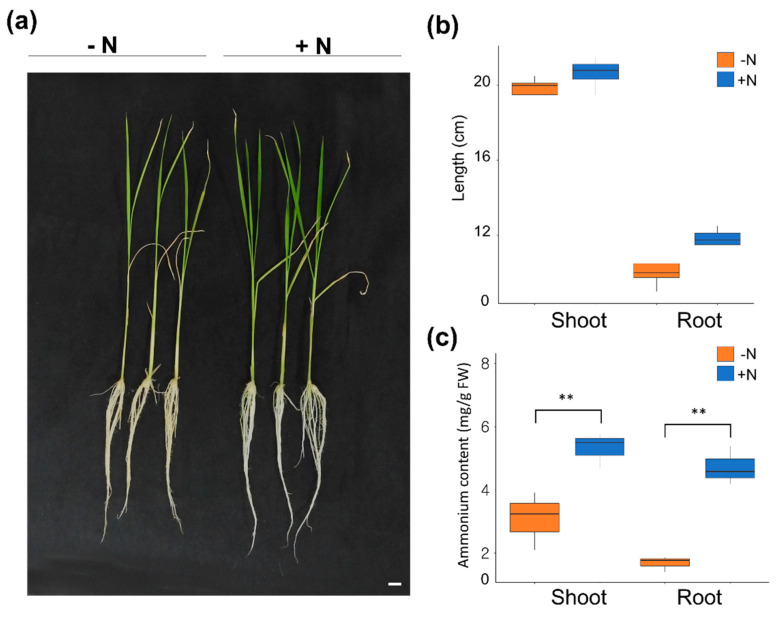
Morphological appearance, quantitative measurement for length, and ammonium measurement of the seedlings under different N conditions in rice. (**a**) Morphology of “*Dongjin*” seedlings. Pre-germinated seedlings were treated with an N-deficient solution for 3 days after growing for 14 days in distilled water. Subsequently, half of those seedlings were transferred to an N-deficient solution, and the other half were transferred to an N-sufficient solution for 48 h. Comparison of the length (**b**) and the ammonium measurement (**c**) of roots and shoots under different N conditions (i.e., –N, N-deficient solution treated for 48 h with 0.5 mM (NH_4_)_2_SO_4_; +N, N-sufficient solution treated for 48 h). This experiment was repeated three times (*n* = 7 for each condition, with n indicating independent experiments in each condition). **, *p*-value < 0.01, based on a *t*-test.

**Figure 3 ijms-22-03243-f003:**
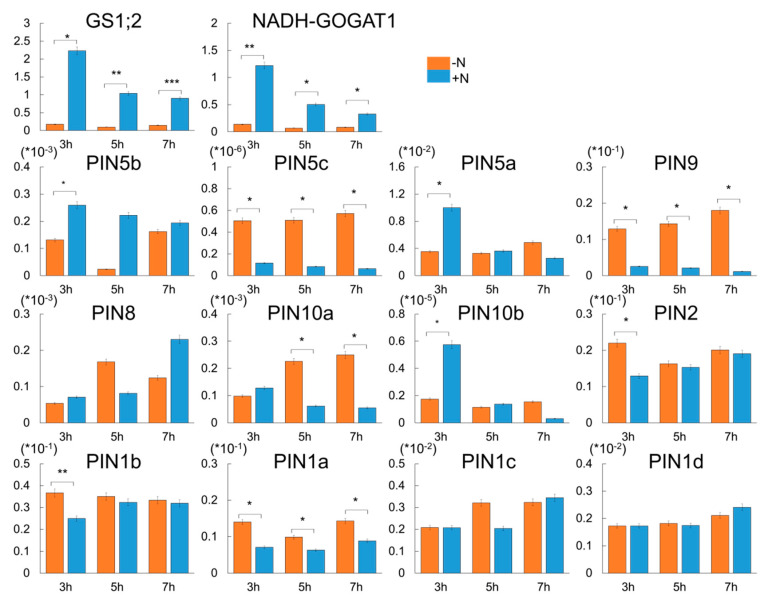
Expression analysis of 12 *OsPIN*-family genes in N-treated roots using qRT–PCR. Rice (*Oryza sativa L. cv. Dongjin*) seeds were grown in either water or N-depleted hydroponic culture medium for 17 days. Half of the seedlings continued to be cultured under the same condition, whereas the remaining seedlings were transferred to N-sufficient medium (0.5 mM (NH_4_)_2_SO_4_) and incubated for up to 7 h. X-axis, sampling time used for qPCR analysis; Y-axis, relative expression level to that of *OsUbi5*. ***, *p*-value < 0.001, **, *p*-value < 0.01, *, *p*-value < 0.05, based on a *t*-test.

**Figure 4 ijms-22-03243-f004:**
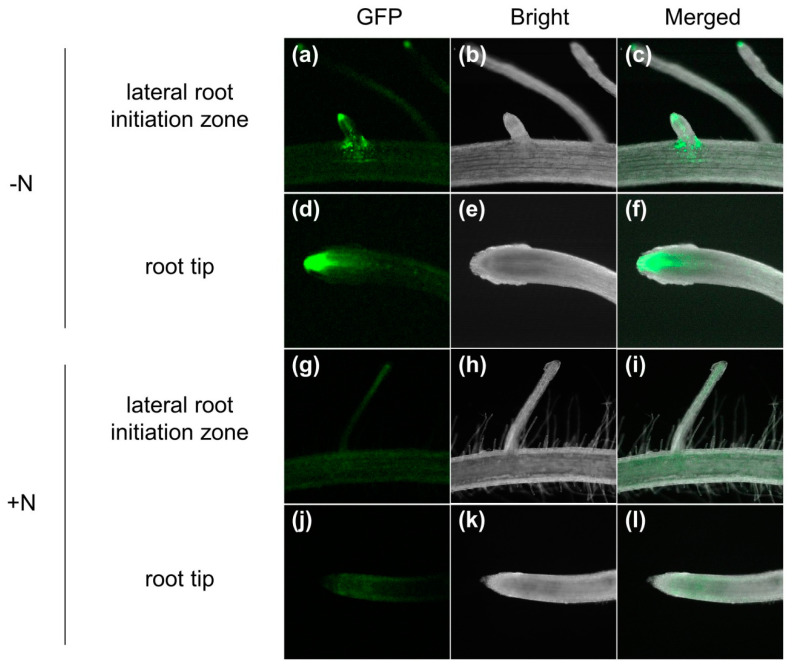
Confocal images of DR5 GFP expression in rice roots. Pre-germinated seedlings were treated with an N-deficient solution (0 (NH_4_)_2_SO_4_) for 3 days after growing for 14 days in distilled water. Subsequently, half of those seedlings were transferred to an N-deficient solution, and the other half were transferred to an N-sufficient solution. Lateral root initiation zone, under N-deficient solution and N-sufficient solution (0.5 mM (NH_4_)_2_SO_4_) (**a**–**f**); root tip, under N-deficient solution and N-sufficient solution (**g**–**l**). –N, roots treated with the N-deficient solution (**a**–**c**,**g**–**i**); +N, roots treated with the N-sufficient solution (**d**–**f**,**j**–**l**) for 48 h.

**Figure 5 ijms-22-03243-f005:**
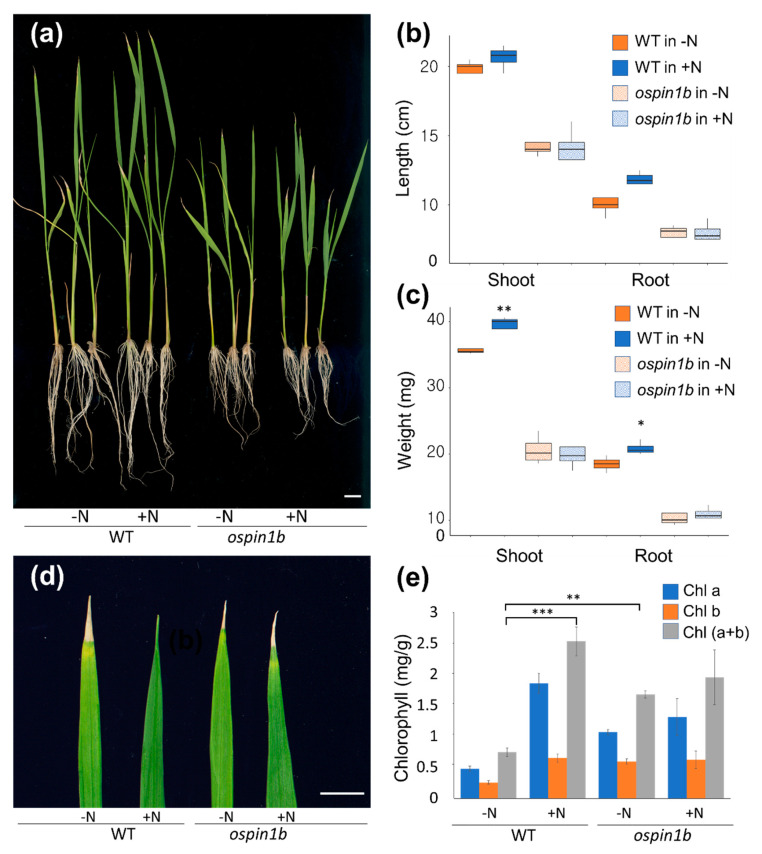
Morphological appearance, quantitative measurement for length and weight, and chlorophyll content measurement of the seedlings between WT and *ospin1b* plants under different N conditions. (**a**) Morphology of WT and *ospin1b* plants at the seedling stage in response to ammonium supplementation after long-term N-deficiency. Pre-germinated seedlings were treated with an N-deficient solution (0 mM (NH4)2SO4) for 3 days after growing for 14 days in distilled water. Subsequently, half of these seedlings were transferred to an N-deficient solution, and the other half were transferred to an N-sufficient solution (0.5 mM (NH4)2SO4). Comparison of the length (**b**) and dry weight (**c**) of roots and shoots between WT and *ospin1b* plants under different N solutions (–N, N-deficient solution; +N, N-sufficient solution) for 48 h. Comparison of the leaf color (**d**) and chlorophyll (**e**) of shoots between WT and *ospin1b* plants under different N solutions (–N, N-deficient solution; +N, N-sufficient solution). Nitrogen was treated for 48 h. This experiment was repeated three times (*n* = 7 for each condition in WT and *ospin1b* plants). ***, *p*-value < 0.001, **, *p*-value < 0.01, *, *p* -value < 0.05, based on a *t*-test.

**Figure 6 ijms-22-03243-f006:**
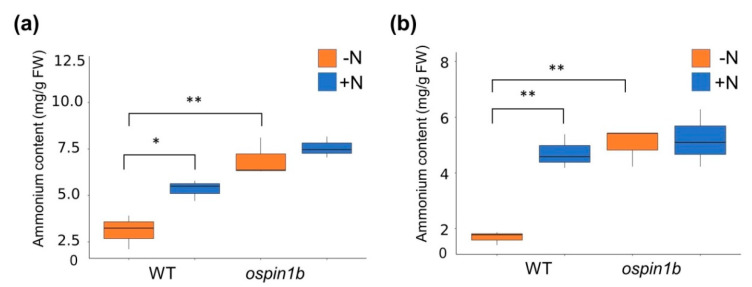
Ammonium content measurement of the seedlings between WT and *ospin1b* plants under different N-conditions. (**a**) The Ammonium content measurement between WT and *ospin1b* of shoot (**a**) and root (**b**) of seedlings between WT and *ospin1b* plants under different N-conditions. Pre-germinated seedlings were treated with a N-deficient solution (0 mM (NH_4_)_2_SO_4_) for 3 days after growing for 14 days in distilled water. Subsequently, half of these seedlings were transferred to an N-deficient solution, and the other half were transferred to an N-sufficient solution (0.5 mM (NH_4_)_2_SO_4_) (–N, N-deficient solution; +N, N-sufficient solution). Nitrogen was treated for 48 h. This experiment was repeated three times (*n* = 7 for each condition in WT and *ospin1b* plants). **, *p* -value < 0.01, *, *p* -value < 0.05, based on a *t*-test.

**Figure 7 ijms-22-03243-f007:**
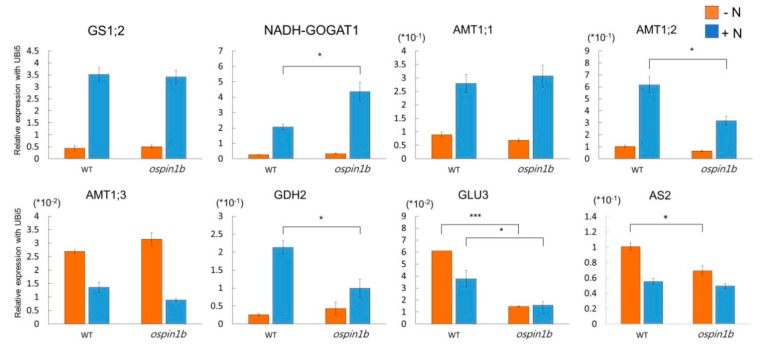
Expression patterns of ammonium-responsive genes between *ospin1b* and WT plants under different N-conditions. The data show the expression patterns of N-assimilation-pathway genes between *ospin1b* and WT plants under different N-conditions using qRT–PCR. The relative expression level to that of the *rice ubiquitin 5* gene (*OsUbi5, LOC_Os01g22490*) was used for the expression analysis. All experiments were repeated using three biological replicates and each replicate was pooled with 5 samples. ***, *p*-value < 0.001, *, *p*-value < 0.05, based on a *t*-test.

**Figure 8 ijms-22-03243-f008:**
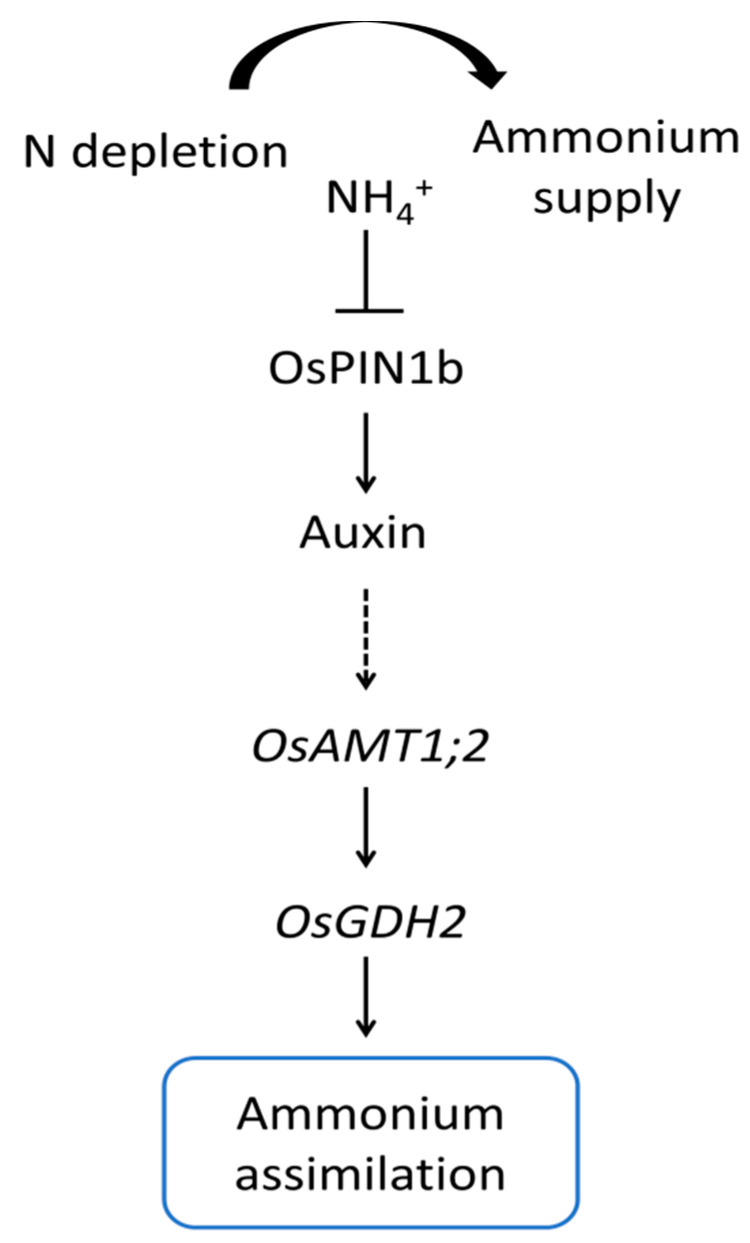
Working model of the function of *OsPIN1b* associated with N-assimilation-pathway genes in rice. Arrows = a positive regulatory action; lines ending in a flat line = a negative regulatory action; dotted line = expected regulatory action.

## Data Availability

Not available.

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
