# Peer review of "Rice PIN Auxin Efflux Carriers Modulate the Nitrogen Response in a Changing Nitrogen Growth Environment"

_ijms, 2021, doi:10.3390/ijms22063243_

Round 1

Reviewer 1 Report

Reviewed manuscript “Rice PIN Auxin Efflux Carriers Modulate the Nitrogen Response in a Changing Nitrogen Growth Environment” is improved version. Authors paid attention for previous remarks. The new manuscript version is supplemented by additional experimental data (ammonium content, chlorophyll content). Figures are rearranged and increased in number. Discussion is expanded.

Nevertheless important remarks are still appeared.

  1. It is mentioned that “auxin signaling” genes are increased in transcription under ammonium treatment as well as transporters genes, but none of them were specified in the text or in final scheme
  2. In Figure 3, relative transcription level of genes encoding PINs family representatives is presented. According to the data PIN 5c, 9 and 10a showed increased expression at N-deprivation. Nonetheless authors used pin 1b in the study instead and gave no clear explanation why so in the text.
  3. Still there is a question about normalization of gene expression to OsUbi5. The experiments are provided with plants differing in growth intensity and under different intensity of stress, thus it is very important to be sure that reference gene is keeping stable expression. It would be useful to add these results to the supplementary.

I would recommend authors to work with manuscript text more time.

Authors need to pay attention also to reference list as well.

Taken together, the manuscript fits the Journal requirements and it could be accepted after minor revision

Author Response

1. It is mentioned that “auxin signaling” genes are increased in transcription under ammonium treatment as well as transporters genes, but none of them were specified in the text or in final scheme.

Response: To address this comment, we revised contents in Pages 1, 11, and 12 and revised Figure 8.

In Page 1, we revised contents in lines 16-17.

In Page 11, we revised contents in lines 350-351. Here, we specified the auxin signaling genes.

In Page 12, we revised the contents in 432. And also we revised Figure 8 by removing "signal".

2. In Figure 3, relative transcription level of genes encoding PINsfamily representatives is presented. According to the data PIN 5c, 9 and 10a showed increased expression at N-deprivation. Nonetheless authors used pin 1b in the study instead and gave no clear explanation why so in the text.

Response: We have only T-DNA insertional lines for OsPIN1b among OsPINs and it was confirmed that the loss of function showed a significant decrease in auxin transporter activity in a previous study [1]. Relating content is prepared in lines 234-235.

[1] Sun, H.; Tao, J.; Bi, Y.; Hou, M.; Lou, J.; Chen, X.; Zhang, X.; Luo, L.; Xie, X.; Yoneyama, K.; Zhao, Q.; Xu, G.; Zhang, Y., OsPIN1b is Involved in Rice Seminal Root Elongation by Regulating Root Apical Meristem Activity in Response to Low Nitrogen and Phosphate. Sci. Rep. 2018, 8, 13014.

3. Still there is a question about normalization of gene expression to OsUbi5. The experiments are provided with plants differing in growth intensity and under different intensity of stress, thus it is very important to be sure that reference gene is keeping stable expression. It would be useful to add these results to the supplementary.

Response: Thank you for your valuable comments. We agree with your opinion.  In order to obtain accurate and reliable gene expression results, careful selection of control genes that exhibit very uniform expression in living organisms at different development stages and under different environmental conditions is important. The reason why we used UBQ5 as a reference gene was that OsUBQ5 was stably expressed in all tissues and plant development stages and at various nitrogen doses in the following three papers. Therefore, we used it as a reference for all qRT-PCR.

  1. Jain, M., Nijhawan, A., Tyagi, A. K., & Khurana, J. P. (2006). Validation of housekeeping genes as internal control for studying gene expression in rice by quantitative real-time PCR. Biochemical and biophysical research communications345(2), 646-651.
  2. Auler, P. A., Benitez, L. C., do Amaral, M. N., Vighi, I. L., dos Santos Rodrigues, G., da Maia, L. C., & Braga, E. J. B. (2017). Evaluation of stability and validation of reference genes for RT-qPCR expression studies in rice plants under water deficit. Journal of applied genetics58(2), 163-177.
  3. Benemann, D. P., Nohato, A. M., Vargas, L., Avila, L. A., & Agostinetto, D. (2017). Identification and validation of reference genes for the normalization in real-time RT-qPCR on rice and red rice in competition, under different nitrogen doses. Planta Daninha35.

We also check the mistakes throughout the text with references. The some changes were marked with red colored letters. 

Reviewer 2 Report

I think that the authors have made all the required corrections. In this form the article is suitable for publication in IJMS

Author Response

Thanks for your help.

This manuscript is a resubmission of an earlier submission. The following is a list of the peer review reports and author responses from that submission.

Round 1

Reviewer 1 Report

The manuscript “Rice PIN Auxin Efflux Carriers Modulate the Nitrogen Response in a Changing Nitrogen Growth Environment” aimed to reveal probable crosstalk between nitrogen deficiency and auxin signaling. It represented newly obtained data on rice as a model crop as an alternative to Arabidopsis. Special interest is paid to encoding of PINs transporters. Authors provided analyze of OsPIN-family genes expression data and focused on alteration in the expression of different PINs in wt and mutant OsPIN1b at different N supply. Thus the manuscript presents important data obtained by several approaches and fully corresponds to the journal scope.

Nevertheless important remarks are necessary.

  1. “Introduction” is mostly described differences in structure and possible functions of representatives of PINs transporters. But it has to be strengthened by the additional information about mechanisms of N supply and metabolization. The importance of different forms of N uptake and specific signaling are required.
  2. The first paragraph of “Results” discovers rice PIN family members. But I couldn’t agree that obtained data somehow deal with its function. Authors did not test transport of auxin exactly.
  3. Expression of genes of the interest was normalized to OsUbi5 (Fig. 3). But according to Fig.6 the expression of “reference” gene varied significantly. Why OsUbi5 was chosen?
  4. Additional information is needed for Fig.6. What means “All experiments were repeated tree times using 5 biological replicates”.
  5. The part “Discussion” is too short. Authors do not pay attention to the role of PIN2 and PIN5 in NH4+-mediated inhibition of root elongation and agraviotropic reaction.

I would recommend authors to work with manuscript text because of some controversial expressions. For example:

line 135-136 – PIN is a transporter, not signaling component.

Authors need to pay attention also to reference list as well.

Taken together, the manuscript fits the Journal requirements and it could be accepted after major revision.

Author Response

The manuscript “Rice PIN Auxin Efflux Carriers Modulate the Nitrogen Response in a Changing Nitrogen Growth Environment” aimed to reveal probable crosstalk between nitrogen deficiency and auxin signaling. It represented newly obtained data on rice as a model crop as an alternative to Arabidopsis. Special interest is paid to encoding of PINs transporters. Authors provided analyze of OsPIN-family genes expression data and focused on alteration in the expression of different PINs in wt and mutant OsPIN1b at different N supply. Thus the manuscript presents important data obtained by several approaches and fully corresponds to the journal scope.

Nevertheless important remarks are necessary.

1. “Introduction” is mostly described differences in structure and possible functions of representatives of PINs transporters. But it has to be strengthened by the additional information about mechanisms of N supply and metabolization. The importance of different forms of N uptake and specific signaling are required.

Response: Page 2, line 52-74. We added the information on the mechanisms of N supply and metabolization.

2. The first paragraph of “Results” discovers rice PIN family members. But I couldn’t agree that obtained data somehow deal with its function. Authors did not test transport of auxin exactly.

Response: To address this comment, we added the contents on the previous study which OsPIN 1b functions as an auxin transporter (lines 291-302).

3. Expression of genes of the interest was normalized to OsUbi5 (Fig. 3). But according to Fig.6 the expression of “reference” gene varied significantly. Why OsUbi5 was chosen?

Response:  Sorry for making confusion on the data, we make similar data for figure 6 with figure 3 and to explain the mechanism regulated by ospin1b, we used heatmap data and the original data in Figure Sx. In general, OsUbi5 has highly stable expression levels across the developmental/anatomical samples in rice, and there are many reports that OsUbi5 is used as an internal control including following reference.

- Bevitori, R., Oliveira, M. B., Sa, M. F. G. D., Lanna, A. C., Da Silveira, R. D., & Silva, S. P. D. (2014). Selection of optimized candidate reference genes for qRT-PCR normalization in rice (Oryza sativa L.) during Magnaporthe oryzae infection and drought.

4. Additional information is needed for Fig.6. What means “All experiments were repeated tree times using 5 biological replicates”.

Response: All experiments were repeated using three biological replicates and each replicate was pooled with 5 samples.

5. The part “Discussion” is too short. Authors do not pay attention to the role of PIN2 and PIN5 in NH4+-mediated inhibition of root elongation and agraviotropic reaction.

Response: We added the suggested content in discussion part (lines 325-338).

6. I would recommend authors to work with manuscript text because of some controversial expressions. For example:

line 135-136 – PIN is a transporter, not signaling component.

Response: As suggested, we revised (lines 158-159).

7. Authors need to pay attention also to reference list as well.

Response: As suggested, we revised it by changing abbreviation of journal name.

Reviewer 2 Report

Auxins play an essential role in the plant physiology, thus topic of presented manuscript is "hot" and important for scientific community. Also rice, as an important agricultural plant  is much better model than Arabidopsis. Authors presented well designed research and interesting results:

  1. An ammonium supplementation reduced the DR5::VENUS signal compared with that observed in the N-deficient condition.
  2. The decreased expression patterns of  OsPIN genes in the presence of the ammonium-dependent response to N deficiency.
  3. The ospin1b mutant showed an insensitive phenotype in the ammonium-dependent response to N deficiency and disturbances in the regulation of several N-assimilation genes.

I have only one small suggestion. Authors should include conclusion to summarize the most important achievements of ms.

Author Response

1. I have only one small suggestion. Authors should include conclusion to summarize the most important achievements of ms.

Response: we added the conclusion part in lines 379-385.